# A Review of Barbed Sutures—Evolution, Applications and Clinical Significance

**DOI:** 10.3390/bioengineering10040419

**Published:** 2023-03-27

**Authors:** Karuna Nambi Gowri, Martin W. King

**Affiliations:** 1Department of Textile Engineering, Chemistry and Sciences, Wilson College of Textiles, North Carolina State University, Raleigh, NC 27606, USA; 2College of Textiles, Donghua University, Songjiang District, Shanghai 201620, China

**Keywords:** barbed sutures, evolution, clinical significance, surgical technique, wound closure, tissue approximation, unidirectional and bidirectional barbed sutures

## Abstract

Surgical ligatures are a critical component of any surgical procedure since they are the device that provides immediate post-surgical tissue apposition. There have been several studies to improve the design and use of these wound closure devices for different surgical procedures. Yet, there is no standardized technique or device that can be used for any specific application. Over the last two decades, there has been an increased focus on the innovative surgical sutures known as knotless or barbed sutures, along with studies focusing on their advantages and disadvantages in clinical environments. Barbed sutures were invented to reduce the localized stress on the approximated tissues as well as facilitating the surgical technique and improving the clinical outcome for the patient. This review article discusses how barbed sutures evolved from the first patent published in 1964 and how these barbed sutures influence the surgical outcomes in different procedures ranging from cosmetic surgery to orthopedic surgery performed on both human patients and animals.

## 1. Introduction

Wound closure devices are used to ligate the incision during a surgical intervention as well as maintain tissue approximation for the duration of wound healing [1,2]. Wound closure is usually the final stage of surgical intervention, which can be performed in three stages, primary, secondary and tertiary closures, which are influenced by the amount of available surrounding tissue for closure and the type and depth of the wound. Wound closure devices are specifically designed to close a wound by holding the diseased, injured or incised tissue together with the help of one of the following devices: surgical sutures, staples, surgical zippers, clips, adhesive tape or adhesive strips, tissue adhesives or laser bonding. These devices are widely used to close cutaneous or skin wounds [3,4], and are designed and fabricated from various materials depending on the precise anatomical site and the function of the approximated tissues. These devices can either be fabricated from permanent or biodegradable materials depending on the longevity of their active function in vivo [2,5].

Medical surgical sutures are the most common type of ligature that has been used during surgical procedures for wound closure and tissue approximation [6]. Sutures contribute to the largest percentage of devices used for wound closure, yet there is no standardized method for securing them [1]. Sutures have been used for wound closure for thousands of years. They were mentioned in ancient Egyptian hieroglyphics, which date back as far as 3000 BC [7]. For several centuries people used natural plant and animal materials such as hemp, cotton, silk and material removed from animals such as tendons and arteries. More recently, they started using catgut, which is made by cutting fine monofilament threads from the intestines of sheep (ovine), cows (bovine) and pigs (porcine) [4].

With technological advancements, there has been a drive towards designing and developing wound closure devices that reduce operating times without increasing the risk of wound dehiscence and other complications compared to existing sutures [8]. During the early 1900s, researchers and surgeons were looking into a novel type of suture that required no knots during tissue approximation [1,6]. The knotless suture was designed to overcome the disadvantages of conventional, non-barbed sutures and thereby improve the patient’s surgical outcome. Later in the mid and late 1900s, the concept of the knotless or barbed suture was described in a number of patents published by both scientific researchers and surgeons. Some of these patents are listed in this article, which reviews the various designs for creating projections or barbs that can be applied to surgical sutures.

During the early days in the development of barbed sutures, surgeons were hesitant to implement these sutures in their clinical practice since they were concerned about the safety and efficacy of these wound closure devices. The early use of these sutures focused on shaping and lifting procedures during cosmetic surgery. This was later expanded to include tissue approximations. However, even though these sutures were designed to improve cosmesis and surgical outcomes, there were several questions as to the safety and efficacy of their use for long-term applications [8]. After the approval by the US Food and Drug Administration (FDA) in the year 2005 for the first barbed suture as a wound closure device for soft tissue approximations, surgeons felt more confident to use them as an alternative to conventional sutures. However, their successful application of, for example, bidirectional barbed sutures required a revised clinical procedure in order to achieve improved surgical efficacy, post-surgical healing and patient comfort. Once these advantages were clarified, this led to more surgeons using barbed sutures in their surgeries, specifically for plastic and cosmetic reconstructive procedures.

## 2. History of Barbed Sutures

### 2.1. Problems Associated with Conventional, Non-Barbed Sutures

For more than a century, surgical wound closure has been performed using conventional braided or multifilament sutures where the surgeon is required to tie knots to secure the suture in situ. Figure 1 represents the three different suture structures that are currently being used for wound closure. Both smooth monofilament (Figure 1A) and braided multifilament structures (Figure 1B) are currently being used for different types of applied stress and tension-free procedures in different anatomical locations. Given the diversity of clinical applications, surgeons have reported that conventional sutures are associated with a risk of complications, such as wound dehiscence, knot slippage, suture rupture and surgical site infections (SSIs). Traditional surgical sutures require knots to be tied where the thread ends and exits the wound. These knots result in an adverse inflammatory reaction and cause friction on the overlying immature scar tissue which can lead to rejection and “spitting” of the suture knot.

One of the major disadvantages is related to knot slippage and suture failure. Since these sutures are composed of smooth monofilaments or braided multifilament yarns, they tend to slip within the knot, where the suture material is exposed to a combination of bending, compressive, tensile and shear stresses. Breakage or failure of the suture is caused by these combined forces acting on the suture material within the knot, which in turn results in wound dehiscence and tissue trauma. Because the knot generates a localized high-stress concentration, the knot becomes the weakest point in the suture line. So, when the knotted sutures are embedded in the dermis at the wound site, they tend to break within the wound itself, resulting in inflammatory and immune reactions within the host body. Sutures are used for closure near or at the dermis, which can result in wound rupture when there is extreme tension leading to patient discomfort and an inflammatory response by the host body [6,8,9]. Along with suture knot failure, the major disadvantage of braided monofilament sutures is that they are prone to attract bacteria that proliferate in the interstices between the filaments where they are shielded from the host’s inflammatory response, which results in wound infection. 

Barbed sutures (Figure 1C) are sutures that have projections along the length of the filament that promote self-anchoring within the surrounding tissues. These barbs help strengthen suture attachment at the wound site, and the operative time is reduced for surgeons who utilize these barbed sutures during their surgical procedures. In contrast to conventional smooth monofilament sutures, barbed sutures are associated with less stress relaxation since the projections are located along the entire length of the filament, thereby resulting in a lower and more uniform retention force distribution. Since barbed sutures are manufactured using monofilaments without any internal pores, there are fewer infections and a reduced risk of an immune response from the host body. Using these barbed sutures in place of conventional sutures is also more economically feasible since barbed sutures are more efficient than smooth filaments, and fewer sutures are required for the same surgical procedure with superior surgical outcomes [10,11,12,13,14].

### 2.2. Barbed Sutures

Suturing and closing bodily tissues after internal surgery is a time-consuming process. However, it is equally important, as with any implantable medical device, that the sutures be inserted in the adjoining tissue meticulously and with precision. Because if there are openings, non-uniformities or discontinuities in the stitching pattern, the risk of wound rupture and dehiscence increases significantly. Unfortunately, when this happens, there is a need to repeat the surgical procedure and cause discomfort to the patient who has previously experienced trauma. So, in order to reduce the risk for both the surgeon and the patient, this barbed suture design with a plurality of projections along the surface makes the operative procedures both more effective and time efficient. The design and concept of barbed sutures have been known for over a century, but they have drawn little attention due to the lack of applied biomedical research and their limited use in clinical practice [15]. Since their approval by the Food and Drug Administration in the US in 2007, barbed sutures have been used in various surgical procedures, primarily for plastic and cosmetic surgery. 

#### 2.2.1. Evolution of Barbed Sutures—Patents Detailing History of Barbed Sutures

The inspiration for the development of barbs on the surface of surgical monofilament sutures came from the structure of a porcupine’s quill, where the barbs are designed with the intention of securely adhering to the surrounding tissue [6]. The use of barbed sutures in medical applications was first mentioned in the mid-1950′s, and the first US patent that described barbed sutures was published by Dr. John Alcamo in 1964, showing the possibility of developing a monofilament suture from fibers, filaments or threads with a rough, bumpy and jagged surface. These surgical sutures are currently known as barbed sutures [16]. In his patent, Dr. Alcamo had reported the design of these sutures (Figure 2) with unidirectional projections, teeth or depressions, which would enable surgeons to use these sutures for tissue approximation without the need to tie a knot. Since his designs were all unidirectional, surgeons had to “double back” to ensure complete closure of an incision or an open wound [3,6,15,16].

In 1967, Dr. Alan McKenzie was granted a UK patent (GB 1091282-A-Sutures) for a bidirectional barbed suture design. He mentioned that the sutures were made of nylon, silver, stainless steel or tantalum filaments. Dr. McKenzie had also mentioned that the bidirectional barbed sutures would facilitate the handling and widen the potential applications for surgeons since they were not required to “double back”, as described earlier by Dr. Alcamo with his unidirectional barbed suture design [3].

In 1978, Taichiro Akiyama claimed in his patent that the projections on the surface needed to be spherical in shape and molded at specific fixed intervals along the length of the suture (Figure 3). Based on Akiyama’s claims, these projections can be molded into three different shapes, such as a cone or bowl. Akiyama also mentioned that this design could be used as a ligature specifically to ligate or close a ruptured blood vessel. He designed these knotless sutures as a replacement for conventional sutures that have a tendency to slip after the suturing procedure [15,17]. 

In 1993, Inbae Yoon published a patent that focused on the design of a surgical closure device that could be used effectively by surgeons performing endoscopic surgery. In his patent, he included a number of designs with tapered barbs but without a needle attached at the end of the suture (Figure 4). Instead of a needle, the designs had a sharp distal end that performs the task of the surgical needle by penetrating the surrounding tissues during surgery [15,18].

Dr. Gregory L. Ruff published two patents on barbed sutures as tissue connectors in 1994 and 2001. He reported that the concept of using barbs eliminates tissue scarring, reduces the risk of tissue necrosis and is associated with shorter operating times during clinical practice [15]. The barbed tissue connector may not be flexible like a traditional suture, but it has sufficient resilience to integrate with surrounding tissue. The design mentioned in the patent in 1994 had a conical array of barbs present on the circumference of the suture (Figure 5) [20]. These barbs may well have been inserted by hand or by an assisting device in order to avoid disrupting the barbs present in the uniformly distributed bidirectional barbed sutures. In the bidirectional barbed suture design mentioned by Dr. Ruff, the barbs are able to yield or collapse when pulled through tissue in the direction of insertion, but they stand up in a rigid configuration when pulled in the other direction since the barbs engage with the surrounding tissue. This results in easy suture insertion and improved anchorage with surrounding tissues. This anchoring mechanism of the barbs makes the sutures more effective and efficient when compared with conventional sutures, which require the time-consuming process of tying knots and a higher risk of scarring, tissue necrosis and wound ischemia [15,20,21].

More recently, in 1999, Harry J. Buncke, the father of modern microsurgery, published a patent in which the innovative concept of barbed sutures was mentioned, along with the technique to cut these barbs or projections along the surface of the suture. In his patent, he also mentioned that barbed sutures could be manufactured through either physical cutting or laser machining. Buncke also introduced various suturing techniques using barbed sutures that can be used for facelifts and brow lifts in cosmetic surgery. For the one-way unidirectional barbed sutures, he recommended using paired sutures so as to maintain anchoring in both directions (Figure 6). Figure 6A represents the unidirectional design of barbed sutures with barbs in the shape of cones, and Figure 6B illustrates the cross-sectional view of the unidirectional barbed suture [22]. 

In 2003, the researchers Steven D. Morency and Jeffery S. Jones described in their patent a type of barbed suture made from a flat filament with a rectangular cross-section and the barbs were created along the lateral edges of the filament (Figure 7). They demonstrated that the barbs have the ability to collapse and flex inwards, permitting easy insertion into tissue. On the other hand, when stress is applied in the opposite direction, the barbs stiffen and resist displacement. In their patent, they mentioned different barb configurations, such as straight and curved, sharp and rounded and convex and concave geometries. As the barbed sutures were made from flat sheet material, they were fabricated by processes such as photo-chemical etching, injection molding, stamping and progressive die cutting [11].

In 2015, Avelar et al. reported a new barbed suture design in which different barbed and non-barbed zones along the monofilament suture were clearly specified and distinguished from each other. They also clearly delineated any change in direction in a bi-directional barbed suture. In the patent, they mentioned the use of a laser system to indicate the difference in features on the surface of these monofilaments, as shown in Figure 8. They described this process of differentiation as a technique to improve the efficiency and functionality of the suture, as well as assist the surgeon to easily identify the different features of the wound closure device during the stressful operating room environment. As the field of surgery moves towards greater automation and the use of indirect robotic arms, the innovation of delineating any change in features and ensuring that this information is an integral part of the delivery system will reduce clinical complications and improve surgical outcomes [23]. 

Ever since the patent published by Dr. Alcamo in 1964, a barbed suture design has evolved through a number of patents published by the United States Patents and Trademark Office, where different researchers have described different barb designs that may be efficient and functional in facilitating various surgical procedures. Nevertheless, not all designs and ideas claimed in the patents have been scaled up and translated into commercial products that can be profitable to the healthcare and medical device industry. Some of the ideas and designs listed above illustrate the evolution of barbed sutures during the past six decades. The barbed suture design published by Dr. Gregory Ruff in his patent published in 1994, which was inspired by the porcupine quill, is the current commercial design known as the Quill^TM^ barbed suture. This and other commercial barbed sutures are described below in Section 2. 

#### 2.2.2. Geometric Design of Barbed Sutures

The geometry of the barb is important to establish the functionality and performance of the particular barbed suture. The reason for this is that different surgical procedures require different barb geometries in order to optimize anchoring with the surrounding tissues. It is important to note that there are many different types of tissue within the human body, and they each have their own structural and organizational characteristics and unique mechanical performance. The two major barb parameters that need to be chosen appropriately are the barb cut depth and barb cut angle (Figure 9).

The length of the barb can be calculated from the cut depth and cut angle of the barb using the following formula: $$L_{C}=\frac{Dc}{\sin\left[180-\theta_{C}\right]}$$
where *L_c_* is the length of the barb cut from the main suture filament, *D_C_* is the cut depth and *θ_c_* is the cut angle, as represented in Figure 9. Depending on the type of surgical procedure, such as incisional or wound closure, the barb parameters will vary in order to meet the mechanical requirements and avoid suture line failure or wound dehiscence. Along with these parameters, the number of barbs present along the suture line is important since it will determine the anchoring capacity of the suture once installed [9,24,25]. 

Barb stiffness is an essential criterion that determines how a barbed suture will perform during and after surgery since the barb is the component that acts as an anchor with the surrounding tissues. A shorter cut depth will result in a stiffer and shorter barb, which may lead to difficulty in penetrating and anchoring surrounding tissue. Surgical procedures that involve tendons and ligaments require barbs with a smaller cut angle and a deeper cut depth in order to achieve efficient anchoring. In the case of dermal surgery, the barbs need to have both a deeper cut depth and a larger cut angle in order to provide better anchoring to the skin tissue, which is flexible, preferentially aligned and thinner than a tendon or ligament [24,25].

The shape, size and morphology of the barbs have a significant influence on the tensile properties and retention strength of barbed sutures. The cut depth is inversely proportional to the tensile strength of the suture. In other words, the deeper the cut depth, the lower the tensile strength of the monofilament suture, because, as more material is removed, the cross-sectional area that can support a tensile load is reduced. In the traditional case of monofilament and multifilament sutures, the tensile strength of the suture is reduced by approximately one-half when securing knots since the knot causes the concentration of many different forces at the same location which then becomes the weakest point in the suture line. It is recommended when using barbed sutures to follow a sinusoidal suture pathway because a curved suture line causes the barbs to stand out and generate improved anchoring with the surrounding tissues. The undulations in the suture line also impart elasticity and reduce the risk of suture breakage, which is a major concern for traditional smooth monofilament and multifilament sutures that require knotting. The barbs or projections are located along the whole length of the suture, which means that barbed sutures do not migrate to the zone of maximum tension, which often occurs during and after surgery with traditional knotted sutures, thereby reducing the efficiency and functionality of open hernia repair and generating longer scars in vivo [24,26,27,28].

## 3. Commercial Barbed Sutures and Their Clinical Significance

### 3.1. Commercial Barbed Sutures

The commercial barbed sutures that are available on today’s market are the STRATAFIX^TM^ (Ethicon, Somerville, NJ, USA), the Quill^TM^ barbed suture (Corza Medical, Westwood, MA, USA) and the V-Loc^TM^ suture (Medtronic, New Haven, CT, USA) [29]. 

#### 3.1.1. STRATAFIX^TM^ Knotless Wound Closure Device

STRATAFIX^TM^ is one of the commercially available knotless tissue control and wound closure devices that is being manufactured for use by plastic and cosmetic surgeons. STRATAFIX^TM^ is the commercial barbed suture manufactured and sold by Ethicon Inc. (Somerville, NJ, USA), one of the leading companies in the wound closure market throughout the world. There are two variations of this device, namely, a symmetrical design and a spiral design. The symmetrical design has barbs like a mirror image on both sides of the suture axis, while the spiral design has a helical array of barbs that protrude around the periphery of the suture, as shown in Figure 10. These knotless wound closure devices are manufactured using both nonabsorbable polymeric monofilaments, such as polypropylene, and resorbable polymers, such as polydioxanone (PDS/PDO) and Monocryl^TM^ (copolymer of glycolide and ε-caprolactone). In addition to the barbs present on the surface, the STRATAFIX^TM^ suture has an antibacterial coating which improves the antimicrobial properties of these barbed sutures and reduces the incidence of surgical site infections (SSIs) [30].

#### 3.1.2. Quill SRS^TM^ Bidirectional Barbed Sutures

Quill SRS^TM^ bidirectional barbed sutures are manufactured by Corza Medical (Westwood, MA). The unique feature of this type of suture design is that the barbs change direction in the middle of the device. Quill barbed sutures are manufactured both as unidirectional and bi-directional barbed sutures. The clinical selection that the surgeon makes depends on the use, the anatomical site and the particular procedure. In the case of unidirectional barbed sutures, the suture is anchored using an adjustable loop on one end, while barbs provide anchoring in the other direction. In contrast, bi-directional barbed sutures have a mid-transition point where the barbs change direction, as shown in Figure 11 [31].

Quill SRS^TM^ barbed sutures are manufactured from monofilaments, such as polydioxanone (PDO), Monoderm^TM^ (a copolymer of polyglycolic acid (PGA) and polycaprolactone (PCL)) and polypropylene (PP). Note that the PP sutures are nonabsorbable monofilaments that are used in tenorrhaphy [3] procedures where a suture with superior chemical and mechanical resistance is required.

#### 3.1.3. V-Loc^TM^ Wound Closure Devices

V-Loc^TM^ is another commercially available barbed suture, which is manufactured by the medical device manufacturing company, Medtronic (New Haven, CT, USA). Their V-Loc^TM^ suture is an innovative wound closure device that consists of unidirectional barbs and an anchor loop at the other end. The V-Loc^TM^ sutures also have a barbed design with a unique dual-angle cut. It has been reported that the suture with a dual-angle cut commanded a higher anchoring force compared to the single-angle cut. Figure 12 shows the difference between the single-angle cut and the dual-angle cut barb design that can be used for a wide variety of surgical procedures [32].

Figure 12 shows that the dual angle cut has a shallower depth and thereby exhibits an improved mechanical performance because the effective cross-sectional area of the suture is increased. V-Loc^TM^ barbed suture wound closure devices are made from different resorbable polymeric monofilaments such as polybutester—a copolymer of glycolic acid and trimethylene carbonate (V-Loc^TM^ 180) and a copolymer of glycolide, dioxanone and trimethylene carbonate (V-Loc^TM^ 90).

Based on recent market research, we determined that the commercial selling price of these barbed sutures is significantly higher when compared to conventional, non-barbed sutures. This reflects the additional cost of fabricating these barbed sutures. Even though the cost of these knotless sutures is higher, they have proven effective since they perform their intended purpose of facilitating surgical handling and reducing the operational time in the OR and providing the patient with an improved surgical outcome.

### 3.2. The Role of Barbed Sutures in Cosmetic and Reconstructive Surgery: Benefits and Complications

Barbed sutures can be used in several surgical situations, such as emergency room procedures, general and thoracic applications, urological surgery, orthopedic and hand applications, obstetric and gynecological procedures, hair restoration and in the majority of plastic, reconstructive and cosmetic applications [33]. 

#### 3.2.1. Cosmetic and Plastic Surgery

Cosmetic and plastic surgical procedures are performed on both healthy individuals and injured patients by using clinical interventions and artistic creativity to improve the patient’s appearance and body image [15,34]. According to the 2020 American Society of Plastic Surgeons (ASPS) statistics report, 15.6 million cosmetic surgeries and about 13.2 million minimally-invasive surgeries are performed each year in the United States [35]. Common cosmetic procedures include rhytidectomy (facelifts), strabismus surgery (correction of eye muscles and droopy eyelids or ptosis repair), rhinoplasty (nose correction), abdominoplasty (tummy-tucks), and breast augmentation and reduction procedures [36]. A suture suspension approach is the most commonly used technique to reconstruct loose and flabby tissue to a tighter and younger-looking appearance during cosmetic surgery. One of the classical suspension techniques is superficial muscle aponeurotic suspension (SMAS), which is also known as rhytidectomy. In this suspension approach, the face muscles are tightened by removing excess skin and fat. An alternative and less invasive technique is the minimal access cranial suspension (MACS), as presented in Figure 13, where the suture is anchored in the deep temporal fascia, and suture loops are inserted to elevate the loose and sagging tissues. This technique involves much less skin excision and is less complicated when compared to the SMAS lift [10,15,37]. Since the development of barbed sutures, these cosmetic surgical procedures have been performed using barbed sutures since they exhibit better surgical outcomes and less scarring compared to conventional sutures. Table 1 lists the various modified sutures that are currently, or have previously been, used in cosmetic and plastic surgical operations.

The use of bidirectional barbed sutures instead of conventional monofilament sutures has obvious advantages of increased speed of placement, easier handling and the freedom from tying knots, as seen in the purse-string effect. Several clinical studies were performed in order to understand the clinical significance and efficacy of barbed sutures, and it was reported by all the surgeons that the use of barbed sutures significantly reduced the closure time when compared to traditional sutures. Procedures such as abdominoplasty, brachioplasty (upper arm lift) and mastopexy (breast lift) require the approximation of three layers of tissue, namely the superficial fascia/deep tissue, deep dermis and superficial dermis. The approximation of these three layers using a traditional suturing technique with conventional sutures requires three separate sutures, but when using bidirectional barbed sutures, only two running sutures are required, and the closure time is reduced by approximately 50%. Figure 14 represents the gradients of tension experienced by the surrounding tissues when a bi-directional barbed suture is deployed during tissue approximation [33].

Innovative barbed surgical sutures play an important role in “lunch-time” facelift procedures since patients prefer the least invasive or non-invasive surgeries. Facial rejuvenation for transforming facial aging has evolved from tension-based procedures such as mini facelifts to various subcutaneous, sub-SMAS and sub-periosteal planes of dissection. Atiyeh et al. investigated the efficacy of barbed sutures used for “lunch-time” facelift surgeries. The “lunch-time” facelift surgery is a procedure that avoids large incisions, significant undermining or substantial recovery time. The study was focused on the different variations of barbed sutures, namely short Aptos^TM^ threads (Tbilisi, Georgia) for Aptos^TM^ lift or Feather lift procedures, long Woffles threads (Kolster Methods Inc., Corona, CA, USA) that were preferred for suspension of sagging tissues and Contour^TM^ threads with unidirectional barbs and an anchor loop on one end, which are known as Feather lift extended Aptos ^TM^ length threads, that were used in different facial rejuvenation surgeries in earlier days when barbed sutures were introduced as an alternative to conventional filaments. It was mentioned that barbed sutures are used in facial aesthetic surgeries, which involve the lifting of brows, midface, lower face and neck. The minor complications that were reported due to the use of barbed sutures were mild facial asymmetry, swelling, erythema, hematoma, slight discomfort and scar formation at the entry and exit sites. When Contour^TM^ threads were used for the facelift procedures, there were noticeable ecchymosis and swelling, which persisted for many days, if not weeks. The study concluded that the use of barbed sutures could be an alternative tool for lifting ptosis tissues, but the preferred use is in open surgical procedures. The choice of barbed sutures requires careful evaluation and acceptance during open rather than laparoscopic procedures [38]. 

Kaminer et al. investigated the long-term efficacy of barbed sutures for minimally invasive thread-lift procedures. They studied the efficacy of Aptos^TM^ threads and the modified Aptos^TM^ threads (Isse Endo Progressive Face Lift Suture^®^) (Tbilisi, Georgia), which are unidirectional polypropylene sutures used in facelift surgical procedures. It was found in this study that patients prefer surgeries that have high-quality results, as found in facelifts, which have less risk and downtime. The study was conducted to understand if the unidirectional and anchored sutures had a long-lasting lifetime in order to support the aging of the skin. The results were in favor of the barbed sutures, as they provide support for a period of 16 months with higher patient satisfaction in contrast to monofilament or braided sutures [39].

Cortez et al. reported a study that investigated complications following the use of barbed sutures for wound closure during plastic surgeries or other cosmetic reconstructions. In their study, they compared the two major commercial barbed sutures available at the time of their study, which were the Quill^TM^ (Corza Medical Inc., Westwood, MA, USA) barbed suture and the V-Loc^TM^ (Medtronic, New Haven, CT, USA) suture. The study involved patients who had undergone breast reconstruction, body contouring and complex free-flap surgical procedures. The study involved both 1-layer closure and 2-layer closure using barbed sutures, which were compared with the same procedure performed using conventional sutures. They reported that the use of barbed sutures reduced the risk of surgical site infections since conventional sutures lead to wound dehiscence, secondary to knot failure and slippage. In this study, they also observed that conventional sutures (mainly braided sutures) have a greater affinity for bacterial adhesion in the interstices resulting in an inflammatory response from the host immune system, unlike the barbed sutures, where there is no specific location along the suture line, which acts like a bacterial attachment site. They summarized that the use of barbed sutures in cosmetic surgeries was advantageous over conventional sutures in terms of surgical risks and outcomes [8]. 

Barbed sutures are used in reconstructive and plastic surgeries since they help surgeons with effective and faster dermal approximation and enable better tissue adherence and support both during and after the procedure. While using barbed sutures, surgeons also found that less suture material was required while closing the dermal tissues, and less operative time was involved in contrast to traditional sutures. 

#### 3.2.2. Obstetric and Gynecological Procedures and Gastrointestinal Surgeries

Barbed sutures have an interesting application in laparoscopic and plastic surgery due to their handling properties. They have been proven effective for performing end-to-end anastomosis ex vivo procedures [40]. One of the most common gynecological procedures where barbed sutures are used is cesarean surgery or the C-section procedure. One other type of procedure which has been gaining attention recently is the laparoscopic procedure with suturing using a laparoscope. While using a laparoscope, the requirement of tying knots would increase the operative time since it is difficult to place knots while performing laparoscopic procedures. The introduction of barbed sutures has improved laparoscopic procedures as they do not require knots to be placed in order to engage with the surrounding tissues. Some surgeries, such as closing the vaginal cuff during total hysterectomy, have benefitted from the use of barbed sutures, where the closure time is reduced by approximately 40–50% when compared to closing the cuff using traditional monofilament or braided sutures [33]. 

Guisto et al. conducted an analysis to compare the suturing techniques and surgical outcomes of two different barbed sutures, namely the unidirectional versus the bidirectional barbed suture, which were, in turn, compared to the traditional monofilament suture. Investigators reviewed earlier studies to find that the use of unidirectional barbed sutures had a higher risk of inflammatory response and were more susceptible to complications than bidirectional barbed sutures. The complications were due to the extended suture with barbs visible and present at the end of the suture point. The researchers mentioned that barbed sutures were safe and effective to be used for end-to-end anastomoses, and the results were coincidental with a lot of other results published by surgeons and researchers on barbed suture efficacy and safety to be used in different clinical applications. Similar to the results published by surgeons previously, investigators of this study also reported that the use of bidirectional barbed sutures was easier, provided less drag and significantly reduced the operating time. In conclusion, they mentioned that barbed sutures could be used in appositional, extra mucosal anastomoses since they promoted anastomotic healing and suture anchoring capacity together with a reduction in surgical operating time [40].

Greenberg and Einarsson reported that the use of bidirectional barbed sutures in laparoscopic and gynecological procedures was beneficial to both surgeons and patients. Barbed sutures help in tissue approximation during laparoscopic surgeries since, during these procedures, tissue approximation is difficult, and the tying of knots is arduous and cumbersome. Even though these surgeries have relied for many years on the tying of traditional suture knots, the introduction of the barbed sutures was ground-breaking and created significantly improved surgical results compared to conventional monofilament sutures. When the post-surgical outcomes associated with clinical ligations were evaluated, both surgeons and patients recognized the benefits of using barbed sutures initially in terms of excellent hemostasis. The use of barbed sutures also reduced tissue trauma since tension is evenly distributed throughout the suture line in contrast to conventional sutures, where the tension is primarily confined to the knots. It was also reported that the bidirectional barbed sutures outshone the same size conventional suture materials in both tensile strength and wound holding capacity as measured in terms of anchoring with the surrounding tissue. Greenberg and Einarsson envisioned that, because of the advantages for both surgeons and patients, the clinical applications for barbed sutures would increase for total laparoscopic hysterectomies, myomectomies and other gynecological procedures [41,42]. Selvest et al. compared the surgical outcomes of vaginal cuff closure during total laparoscopic hysterectomies while using both conventional and barbed sutures. At the end of the comparative study, they concluded that barbed sutures exhibited superior surgical outcomes and performance in comparison to conventional sutures [43].

Demyttenaere et al. studied the advantages and complications of V-Loc^TM^ advanced wound closure devices (Medtronic, New Haven, CT) when used in laparoscopic surgery. In their study, the researchers investigated the post-surgical outcomes of using V-Loc^TM^ sutures in gastrointestinal enterotomy closures. As mentioned earlier, laparoscopic surgery is a surgical technique that involves suturing with limited visualization and thereby requires wound closure devices or sutures that are easier to tie knots both at the incision and at the exit of the suture line. These V-Loc^TM^ sutures are unidirectional barbed sutures that can be utilized for enterotomy closure during different laparoscopic surgeries. It was found that the enterotomy closure time, or anastomotic time, was significantly reduced and faster when V-Loc^TM^ sutures were used for suturing the jejunum, colon and stomach. In conclusion, Demyttenaere et al. reported that barbed sutures provide an efficient and effective alternative to conventional sutures for gastrointestinal surgical procedures [44]. Huang et al. predicted the surgical outcomes of laparoscopic myomectomy when using two commercial sutures, QuillTM (Corza Medical Inc., Westwood, MA, USA) barbed suture and the V-LocTM (Medtronic, New Haven, CT) suture. The authors mentioned that the use of barbed sutures reduced the surgical time and also improved the surgical outcomes of the procedure [45].

Giampaolino et al. studied the outcomes of laparoscopic myomectomy when performed using conventional sutures and the commercial bidirectional barbed sutures STRATAFIX^TM^ (Ethicon Inc., Somerville, NJ, USA). In this study, they compared the surgical outcomes of different types of sutures used during laparoscopic posterior myomectomy. Similar to the other studies, the researchers in this study concluded that the use of barbed sutures reduced the operating time, suturing time and blood loss, and there was no significant difference in postoperative adhesions between the two types of sutures [46].

Barbed sutures are ideal for use during abdominoplasty because progressive tension sutures (PTS) are required to accommodate a drainage catheter. Gutowski et al. studied the safety and efficacy of incorporating barbed sutures during abdominoplasties. In their report, they compared the effects of Quill^TM^ sutures and V-Loc^TM^ sutures against nonabsorbable non-barbed sutures for their biomechanical performance, safety and effective use in different procedures involving abdominoplasties. In the discussion of the biomechanical performance of barbed sutures, they reported that there was no significant difference in the host tissue response, and barbed sutures were able to support the wound healing process by offering strength to the wound during the first post-surgical week when there are greater mechanical forces exerted on the wound. Along with the various advantages of barbed sutures that were reported in a number of clinical reviews, the investigators noted that suture placement of barbed sutures was easily understood and learned by surgeons during their training. Along with body contouring in abdominoplastic surgeries, barbed sutures are also used in arm-lift and thigh-lift procedures. The investigators concluded that barbed sutures are safe and effective to be used for both standard and circumferential abdominoplasty procedures and that they have a similar or superior biomechanical performance when compared with monofilament or braided sutures [47].

Based on the reviews and studies published by surgeons following a number of clinical trials, it is now accepted that barbed sutures are safe, effective and efficient for use in different obstetric and gynecological procedures with acceptable patient comfort and surgeon satisfaction.

#### 3.2.3. Orthopedic Procedures

Orthopedic arthroplasty involving barbed sutures has consistently shown reduced operating times, faster wound closure times and better surgical and postoperative outcomes when compared to conventional sutures [33,48]. Barbed sutures are increasingly being used in orthopedic procedures since they have obvious advantages, including faster tying, more even distribution of retention forces throughout the suture line, and there is no need for complex instruments [33]. 

Johnston et al. studied and reported on the advantages of barbed sutures over conventional sutures for spinal surgical procedures. There is limited evidence reporting the use of barbed sutures for closing incisions during spinal surgeries, but in those few cases, it has been reported that barbed sutures result in faster closure times and improved postoperative outcomes in comparison to using conventional sutures. Barbed sutures have shown promising outcomes when used to treat spine-related disorders such as spinal deformities, spinal infections, trauma, spinal tumors and degenerative spine conditions, such as stenosis and herniated disks. These latter conditions are some of the most prevalent musculoskeletal disorders which definitely require surgical intervention since non-invasive treatments have been reported to have no beneficial effects. 

Johnston’s research group compared the results of surgery using conventional suture materials for suture line closures against the STRATAFIX^TM^ (Ethicon Inc., Somerville, NJ, USA) knotless tissue control device. The post-surgical performance was measured in terms of operating time, duration of postoperative stay, wound complications and readmissions in the unlikely case of any post-surgical incidents. In their study, they emphasized that successful suture line closure is an important factor regarding the postoperative outcome of spinal surgery because it influences healing, surgical-site infections (SSIs), the ability of patients to perform self-care and post-acute care follow up. Johnson et al. reported that barbed sutures had proven efficient for subcutaneous wound closure in spinal surgical interventions. The researchers reported that the STRATAFIX^TM^ knotless barbed suture provided superior performance compared to conventional sutures in terms of operating room time and postoperative outcomes. Having said that, they found no significant difference in suture line complications or readmissions after surgery [48].

More recently, Shah et al. reported on the use of barbed sutures in tenorrhaphy and surgical procedures that repair injured tendons. The use of barbed sutures in tendon repair has been reported to have comparable outcomes with regard to appearance, reduced wound scarring, tissue approximation and the risk of dehiscence. The use of barbed sutures reduced localized tissue trauma with respect to the distribution of retention forces along the entire length of the implanted suture. In this study, the authors reviewed the use of barbed sutures (Quill^TM^ and V-Loc^TM^) in tenorrhaphy compared to conventional non-barbed sutures. Their report was based on the following criteria: maximum load to failure, mode of failure, load to 2 mm gap formation, type of repair, changes in the cross-sectional shape, and the type of repair involved. It was explained that the use of non-barbed sutures that require knots for device placement led to the generation of high stresses at the knots themselves. The presence of knots increases the cross-sectional area of the tendon and increases the frictional resistance within the surrounding tissue. The investigators also performed a maximum pull-out force test on barbed and non-barbed sutures for tendon repair and determined that due to a more uniform distribution of anchoring force throughout the suture line, the barbed sutures exhibited a higher maximum pull-out load compared to traditional sutures. So even though the concept of using barbed sutures for tenorrhaphy is not widely accepted, the authors believe that due to their superior performance, barbed sutures can be used successfully to replace traditional non-barbed sutures for tendon repair [3,49]. 

Wang et al. studied barbed sutures with a symmetric anchoring design and conventional interrupted sutures for total knee arthroplasty (TKA). In particular, they compared the wound closure efficacy and safety of the symmetric anchoring suture (STRATAFIX^TM^ Symmetric PDS^TM^ Plus, Ethicon Inc., Somerville, NJ) with conventional sutures when used in a TKA procedure. From previous meta-analyses, barbed sutures resulted in shorter operating times and improved cost-effectiveness even when there were similar rates of wound complications as with conventional sutures. When interrupted sutures were used for TKA procedures, there was relatively low suture efficacy reported. The use of barbed sutures diminished the risk of complications at the incision, reduced wound dehiscence and lowered the risk of local tissue ischemia and hematomas. Researchers in this study determined the efficacy and safety of barbed sutures using primary and secondary endpoints, which included surgical incision closure time, operating time, total operating room time, length of post-surgical stay, as well as the level of pain and range of motion of the post-surgical knee joint. The symmetrically anchored design of barbed sutures eliminated the need to tie knots at the end of the suture line. These barbed sutures reduced surgical incision closure time and achieved faster arthrotomy closure times as well as fewer complications compared to conventional sutures. These advantages made the use of barbed sutures a desirable option for TKA since TKA procedures require surgeons to quickly close the wound so as to decrease the risk of infections at the surgical site. It was also evident that the use of barbed sutures increased the blood flow since they can achieve a more even distribution of stress which reduces tissue trauma during the surgery and facilitates easier and more extensive joint motion after the surgery [14,50].

Mayet et al. studied the use of barbed sutures in foot and ankle procedures. More specifically, the researchers evaluated the use of the Quill^TM^ (Corza Medical Inc., Westwood, MA, USA) barbed suture as a locking suture for these repair surgeries. In a previous clinical study, Chowdry et al. reported poor wound closure and severe scar formation involved in foot surgery when using the V-Loc 180^TM^ barbed suture (Medtronic, New Haven, CT, USA) [51]. In contrast, Mayet et al. reported that Quill^TM^ barbed sutures in foot surgery achieved better surgical outcomes and lower wound complications. V-Loc^TM^ sutures are unidirectional barbed sutures with a loop at one end and barbs all unidirectionally aligned, whereas the Quill^TM^ sutures are bidirectional barbed sutures. Post-surgical complications have been reported, which include suture extrusion, suture visibility, a severe inflammatory response and infection. These complications are thought to depend on the suture material and its chemical composition. In the case of barbed sutures, such complications are also influenced by the barb geometry as well as the frequency and alignment of the barbs. The limitation of this study is that the barbed sutures were made from different materials, which behaved differently in terms of their degradation and absorption rates within the surrounding host tissue. For this reason, the clinical observations of these barbed sutures cannot be directly compared to other studies. In conclusion, the authors reported that these Quill^TM^ sutures are safe and effective to use in foot and ankle surgeries resulting in both higher patient and surgeon satisfaction [52].

Even though barbed sutures have various advantages over conventional sutures in orthopedic surgeries, the mechanical properties of barbed sutures are lower than conventional sutures. In the case of orthopedic surgeries, the sutures should have higher breaking strength since the stresses applied in these joint locations are higher compared to other locations of the body. Although barbed sutures do not possess the required mechanical strength, they are still being used in joint replacement surgeries as they are easily deployed in intricate areas without the requirement of knots during placement.

## 4. Conclusions

As mentioned previously in the history of barbed sutures, there have been several patents and studies related to the design of barbed sutures, but not all designs have been adopted commercially. After the FDA approval in 2005 to use these barbed sutures for soft tissue approximations, a number of surgeons looked into the safety and efficacy of these barbed sutures by using them in different clinical procedures. As mentioned in clinical studies and reviews by a number of researchers and surgeons, the use of barbed sutures significantly lowered the wound closure time and tissue trauma, along with providing satisfying results to both surgeons and patients. Even though there were complications reported in some cases due to the use of certain types of barbed sutures, on the broad spectrum, it was reported that barbed sutures were safe and efficient in different types of surgical procedures. The major limiting factor is the production rate of barbed sutures. From a current commercial perspective, the manufacturing of barbed sutures is expensive, since it requires skilled technicians and specialized equipment and assemblies to produce consistent barbed sutures. It is also understood that different surgical procedures require different barb geometries in order to arrive at better surgical outcomes [24,25].

Barbed sutures are an innovative and effective alternative to conventional surgical sutures. This is not only because barbed sutures reduce the operating time in the OR, but they also lead to better surgical outcomes with less stress on the approximated tissues. Although barbed sutures have significant advantages over conventional sutures, as reported in several clinical studies, they are not widely used by surgeons due to the fact that barbed sutures have a higher manufacturing cost and command a higher selling price. They also require the surgeon to take training courses to adapt to a more skillful revised surgical procedure. This makes the topic of barbed sutures an interesting and productive area for research with the goal of reducing manufacturing costs, thereby making the device more attractive from both a financial and surgical point of view, as well as providing the patient with faster healing and improved cosmesis. 

## 5. Future Directions

In the future, the fabrication of these barbed sutures can be approached through alternative means of molding, 3D printing or micro-machining so as to satisfy the requirement of different barb geometries and morphologies required for different surgical applications. Lowering the manufacturing cost of barbed sutures by increasing the production rate and reducing the technical errors involved in fabrication will eventually lower the cost of use of barbed sutures and increase the number of surgeons who prefer to use them in place of conventional monofilament or braided multifilament sutures. Barbed sutures lower the operative time and have better surgical outcomes. 

## Figures and Tables

**Figure 1 bioengineering-10-00419-f001:**
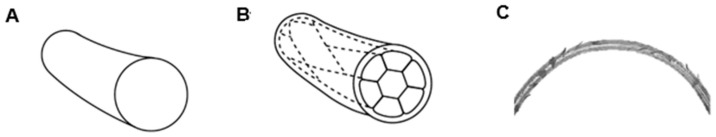
Typical suture threads used in surgical procedures. (**A**) Smooth monofilament suture; (**B**) braided multifilament suture; (**C**) plurality of barbs on the periphery of a monofilament suture.

**Figure 2 bioengineering-10-00419-f002:**
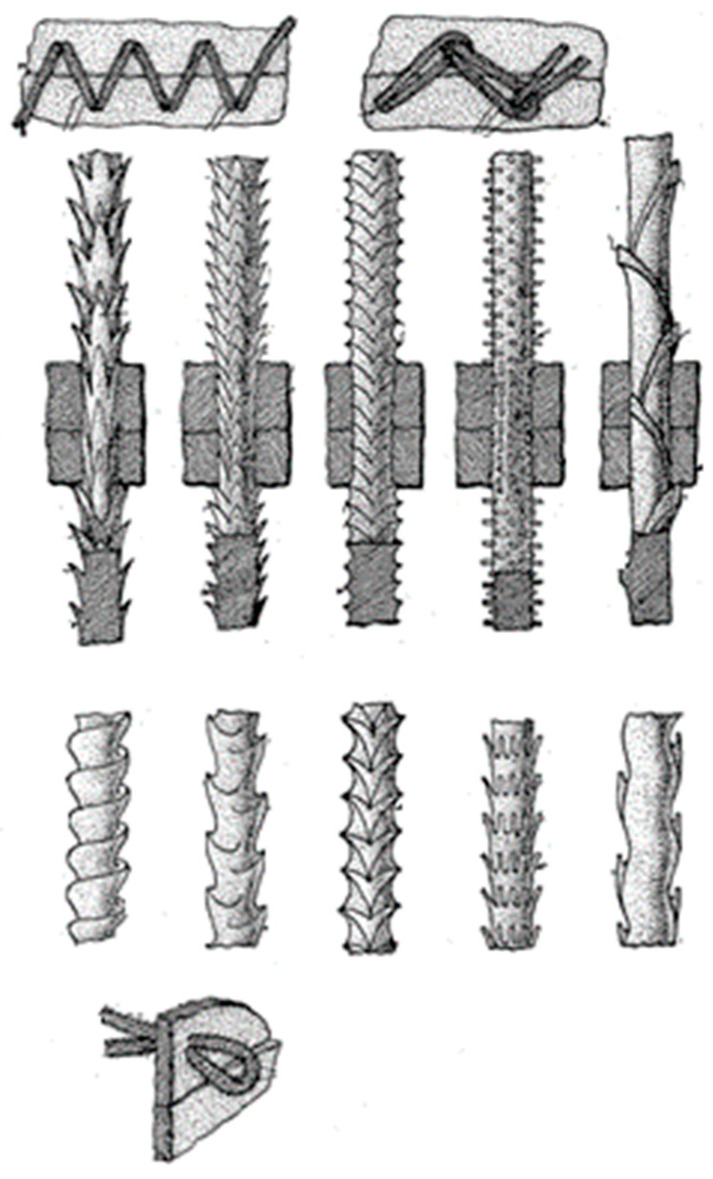
John Alcamo’s surgical suture configurations with unidirectional projections, teeth and/or depressions [16].

**Figure 3 bioengineering-10-00419-f003:**
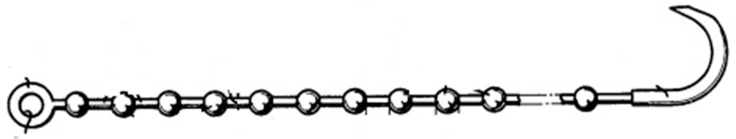
Taichiro Akiyama’s design of a knotless suture with molded spherical projections [17].

**Figure 4 bioengineering-10-00419-f004:**
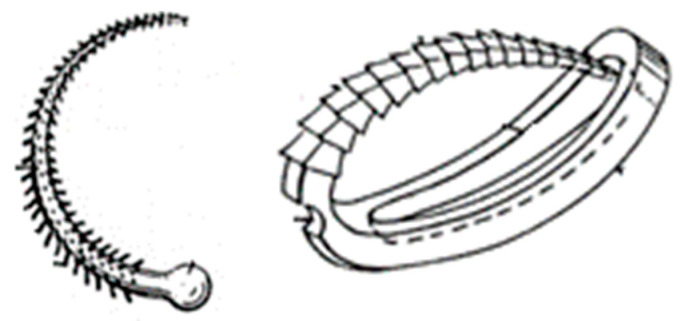
Dr. Yoon’s surgical devices with tapered or whisker-like barbs for endoscopic surgery [18,19].

**Figure 5 bioengineering-10-00419-f005:**
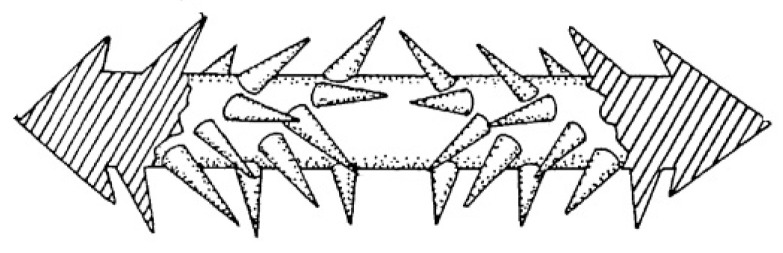
Dr. Gregory Ruff’s conical barb design of a tissue connector [20].

**Figure 6 bioengineering-10-00419-f006:**
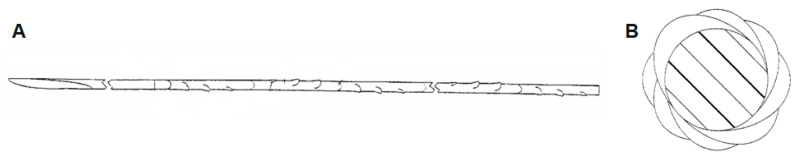
(**A**) Harry Buncke’s unidirectional barbed suture with barbs cut in a conical sequence and (**B**) a cross-sectional view of the barbed suture [22].

**Figure 7 bioengineering-10-00419-f007:**
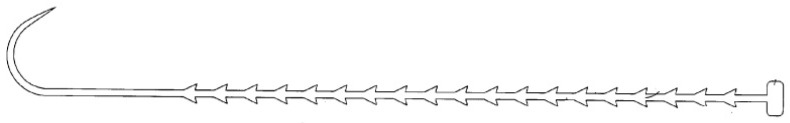
Morency and Jones’ barbed suture, fabricated from a flat sheet with the barbs cut along the lateral edges [11].

**Figure 8 bioengineering-10-00419-f008:**
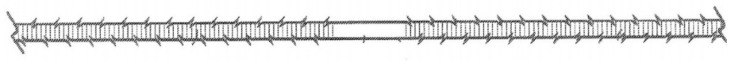
Avelar et al. included laser markings indicating the barbed and non-barbed sections of the barbed suture [23].

**Figure 9 bioengineering-10-00419-f009:**
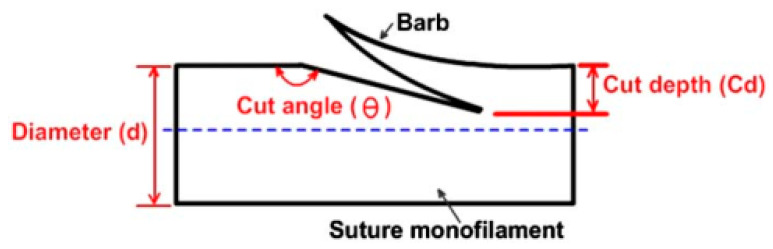
Geometry of a single barb [24].

**Figure 10 bioengineering-10-00419-f010:**
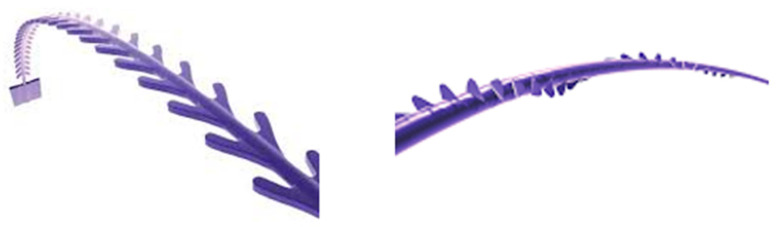
STRATAFIX^TM^ PDS plus knotless tissue control and wound closure devices: **Left**: Symmetrical barbs located on both sides of the suture. **Right**: A spiral array of barbs located around the periphery [30].

**Figure 11 bioengineering-10-00419-f011:**
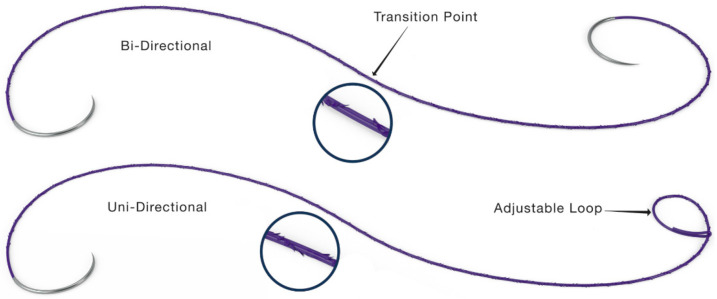
Top: Quill^TM^ bi-directional barbed suture. Bottom: Quill^TM^ unidirectional barbed suture [31].

**Figure 12 bioengineering-10-00419-f012:**
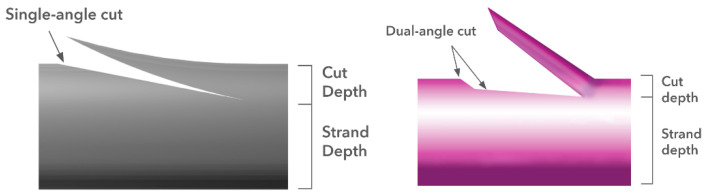
Different barb design for V-Loc^TM^ suture. Left: single angle cut barb. Right: dual angle cut barb [32].

**Figure 13 bioengineering-10-00419-f013:**
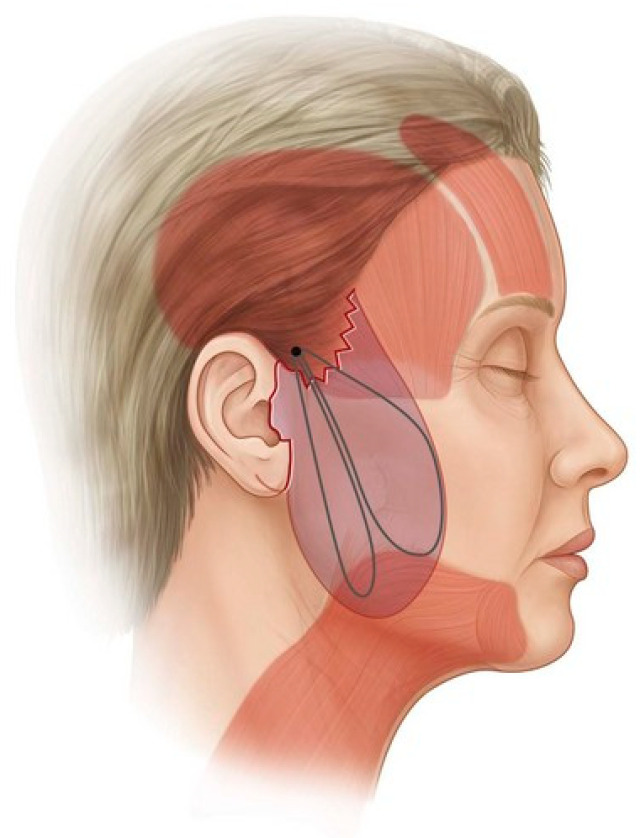
Basic MACS lift where the suture is anchored in deep temporal fascia and sutures are looped in a purse-string fashion [37].

**Figure 14 bioengineering-10-00419-f014:**
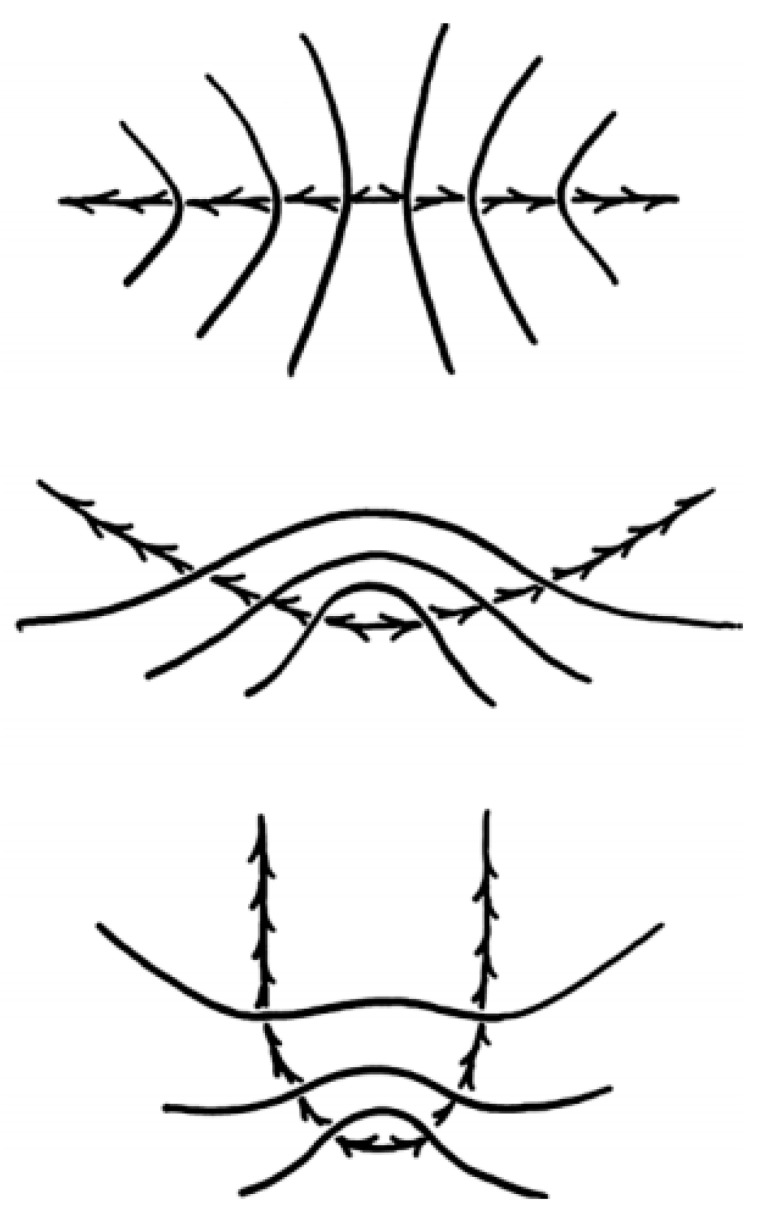
Gradients of tension experienced by the surrounding tissues. Linear compression at the point where the barbs change direction (**top**); Accurate placement imparts a mound by adding a vertical vector to the horizontal vector (**middle**) and U-shaped deployment results in the forces on the barbs being opposed by the bend in the U, rather than by the opposing barbs (**bottom**) [10,33].

**Table 1 bioengineering-10-00419-t001:** Comparison of different available barbed suture devices (Highlighted rows: Currently used commercial barbed sutures available for different surgical procedures).

Suture Name	Description	Suture Placement
Aptos Thread (Kolster Methods, Inc., Anahelm, CA, USA)	Bidirectional, nonabsorbable barbed suture	Free floating
Contour Thread (Surgical Specialities, Reading, PA, USA)	Unidirectional, nonabsorbable, looped or nonlooped barbed suture	Anchored proximally
Isse Endo Progressive Facelift Suture (Kolster Methods, Inc., Anahelm, CA, USA)	Unidirectional, nonabsorbable barbed suture	Anchored proximally
Silhouette Mid-Face Suture (Kolster Methods, Inc., Anahelm, CA, USA)	Nonabsorbable suture material with absorbable knots at 10 mm intervals	Anchored proximally
Woffles Thread (Kolster Methods, Inc., Anahelm, CA, USA)	Bidirectional, nonabsorbable barbed suture doubled in a sling format	Anchored proximally
**V-Loc ^TM^ Wound Closure Device (Medtronic, New Haven, CT, USA)**	**Unidirectional, knotless, absorbable barbed suture**	**Anchored or not anchored**
**Quill^TM^ Knotless Tissue-Closure Device (Corza Medical Inc., Westwood, MA, USA)**	**Absorbable and nonabsorbable, knotless, bidirectional barbed suture with central non-barbed segment**	**Anchored or not anchored**
**STRATAFIX^TM^ (Ethicon Inc., Somerville, NJ, USA)**	**Bidirectional, absorbable barbed suture**	**Anchored or not anchored**

## Data Availability

Not applicable.

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
