# Peer review of "A Review of Barbed Sutures—Evolution, Applications and Clinical Significance"

_bioengineering, 2023, doi:10.3390/bioengineering10040419_

Round 1

Reviewer 1 Report

Dear Authors,

You have undertaken an ambitious, interesting and timely research topic.

I approached reading your publication with great hope and interest.

During the study of the manuscript, significant doubts arose in me regarding its structure and individual threads.

1.     The publication is a narrative review, which should be emphasized both in the title and in the text.

2.     Even a narrative review constituting a certain subjective attitude of the authors to the scientific issues they raise must comply with certain rigors in order to preserve the scientific overtones of the publication, 

3.     I, on the other hand, did not find in your manuscript a clear purpose of publication and criteria for the selection of references.

4.     When you describe the advantages of barbed sutures, you rely on the claims contained in the patent descriptions and not on the results of experimental research. It is not entirely credible as the claims contained in patent descriptions are often only a description of the patent author's beliefs and not an experimentally proven fact (references 14-22, lines 151-188, 215-226)

5.     There is only one subsection 1.2.1.1 in section 1.2.1. which does not justify its specification, but rather prompts a change of the title of chapter 1.2.1.

6.     The study is very broad but scientifically very superficial. Reading it has the impression of many repetitions. Personally, I would recommend shortening the text by two thirds, leaving the reader with the most important achievements of barbered suture. The whole publication deserves to be published in the form of a monograph, not a position in a scientific journal.

7.     In chapter 2.1. you list only 3 commercial barbed sutures, while in table 1 chapter 2.2.1. you give as many as 8 of them. These are inconsistent data.

8.     In its current form, the publication sounds more like an advertisement for barbed sutures than a review. You clearly euphorize this form of tissue stabilization, depreciating the role and exposing the disadvantages of other methods. Despite the large volume of the work, You did not include information about the disadvantages, complications and limitations of barbered sutures.

9.     The work has an incoherent, disturbed layout and, in my opinion, in the presented form does not meet the requirements for objective scientific work. 

10.  The discussion contains only 2 references, which are rather the basis of the list of praise for the method described in this chapter than a discussion of the pros and cons of the method or its challenges and directions of development

11.  Conclusions are rather a continuation of the list of praise for the method and authors' beliefs not supported by the content of the work, and not conclusions in the sense of a scientific work. The conclusions are too broad.

In its current form, I find it difficult to recommend your manuscript for publication.

Best regards

Author Response

Thank you for providing your valuable comments on the manuscript.

- The publication is a narrative review, which should be emphasized both in the title and in the text.

The title has been edited to emphasize that the manuscript is a narrative review on the topic.

- Even a narrative review constituting a certain subjective attitude of the authors to the scientific issues they raise must comply with certain rigors in order to preserve the scientific overtones of the publication,

The authors have reviewed the scientific issues of barbed sutures and interpreted the literature findings with rigor and objectivity.

- I, on the other hand, did not find in your manuscript a clear purpose of publication and criteria for the selection of references.

The purpose of the narrative review is described in the introduction section and the references were selected using the keywords: barbed suture, wound closure, clinical significance

- When you describe the advantages of barbed sutures, you rely on the claims contained in the patent descriptions and not on the results of experimental research. It is not entirely credible as the claims contained in patent descriptions are often only a description of the patent author's beliefs and not an experimentally proven fact (references 14-22, lines 151-188, 215-226)

In this section, we wanted to briefly design and detail out the barbed suture designs that were mentioned from the very first patent that was published in the year 1964. This section was written in the view to mention how the design of barbed sutures evolved over the years until the design that is commercially being manufactured by the commercial surgical suture manufacturing companies.

- There is only one subsection 1.2.1.1 in section 1.2.1. which does not justify its specification, but rather prompts a change of the title of chapter 1.2.1.

We have made the change by removing the subsection 1.2.1.1 and changed the title of the chapter 1.2.1.

- The study is very broad but scientifically very superficial. Reading it has the impression of many repetitions. Personally, I would recommend shortening the text by two thirds, leaving the reader with the most important achievements of barbered suture. The whole publication deserves to be published in the form of a monograph, not a position in a scientific journal.

We did try our best to shorten the manuscript but we did not want to miss out on some of the important contents related to barbed sutures evolution and their clinical significance and that is the reason that the manuscript is longer than expected.

- In chapter 2.1. you list only 3 commercial barbed sutures, while in table 1 chapter 2.2.1. you give as many as 8 of them. These are inconsistent data.

The currently available barbed sutures that are commercially present in the market are mentioned in section 2.1 along with the design of the barbed sutures available. In the section 2.2.1, the table lists all the previously used barbed sutures in surgical procedures along with the commercial ones (highlighted). In order to make it better to perceive by the readers, we have highlighted the current barbed sutures that are commercially available.

- In its current form, the publication sounds more like an advertisement for barbed sutures than a review. You clearly euphorize this form of tissue stabilization, depreciating the role and exposing the disadvantages of other methods. Despite the large volume of the work, You did not include information about the disadvantages, complications and limitations of barbered sutures.

The disadvantages, complications and limitations of barbed sutures in different clinical studies are mentioned in lines 440-450, 470-475, 530-540,  628-640 as part of sections 3.2.1, 3.2.2 and 3.2.3 (clinical significance of barbed sutures in different surgical procedures).

- The work has an incoherent, disturbed layout and, in my opinion, in the presented form does not meet the requirements for objective scientific work. 

In response to this comment, the authors need guidance as to how the topic should be presented in a coherent, logical and objective manner.

- The discussion contains only 2 references, which are rather the basis of the list of praise for the method described in this chapter than a discussion of the pros and cons of the method or its challenges and directions of development

The discussion section has been re-edited to the conclusion section in the manuscript since the discussions are made as part of the clinical significance section 3.2.

- Conclusions are rather a continuation of the list of praise for the method and authors' beliefs not supported by the content of the work, and not conclusions in the sense of a scientific work. The conclusions are too broad.

We have made the necessary changes and edits to the conclusion section and have split the body into conclusions and future directions.

Reviewer 2 Report

This is a comprehensive review of barbed sutures and their use in multiple different fields of surgery. It is concise, interesting, and a more general review of the topic than reports previously published.

Some of the figures are taken directly from referenced text with little to no editing. This makes them difficult to understand as in figure 1 where two of the subpanels are labeled as “B” and as “C”. Some of the sub-labels have a pilcrow that is unexplained in figures 1 and 6.

Other figures maintain patent labels marked as numbers or letters as in figures 2,3,4 presumably explained in the original document, but are not addressed in this manuscript. These excess labels make figure interpretation difficult when trying to interpret within the scope of this manuscript.

Figures 10, 11, and 12 appear to be from package inserts which are not referenced. Information regarding the products in section 2 including Stratafix, Quill, V-Lock is only referenced to citation 28, which is another review. 

Author Response

Thank you for providing your valuable comments on this review manuscript.

- Some of the figures are taken directly from referenced text with little to no editing. This makes them difficult to understand as in figure 1 where two of the subpanels are labeled as “B” and as “C”. Some of the sub-labels have a pilcrow that is unexplained in figures 1 and 6.

We have mentioned the figure labels within the manuscript's body and made the necessary edits to have more clear representation of the figures within the body. We have also removed the pilcrows from the figure labels in figures 1 and 6 which was added as a mistake.

- Other figures maintain patent labels marked as numbers or letters as in figures 2,3,4 presumably explained in the original document, but are not addressed in this manuscript. These excess labels make figure interpretation difficult when trying to interpret within the scope of this manuscript.

We have edited the figures by removing the unwanted labels and texts involved within the figures. 

- Figures 10, 11, and 12 appear to be from package inserts which are not referenced. Information regarding the products in section 2 including Stratafix, Quill, V-Lock is only referenced to citation 28, which is another review.

We have referenced the figure 10,11 and 12 in the manuscript with their respective catalog references and also mentioned in the body of the manuscript.

Reviewer 3 Report

Dear authors

This manuscript reviewed the evolution, application and clinical significance of the barbed sutures. The authors discussed how barbed sutures evolved  from the first patient and how these barbed sutures influence the surgical outcomes in different procedures ranging from cosmetic surgery to orthopedic surgery performed on both human patients and animals.   

This review is very interesting and the contents will be able to give the precious informations to clinicians in various fields. I recommend that the manuscript has enough potential for the publication in the Journal.

I think that the content of the manuscript is too long, please make it shorter.

Best regards

Author Response

Thank you for providing your valuable comments on the manuscript.

We did try our best to shorten the manuscript but we did not want to miss out on some of the important contents related to barbed sutures evolution and their clinical significance and that is the reason that the manuscript is longer than expected.

Reviewer 4 Report

The review article by Karuna Nambi Gowri was well written and summarized the Barbed sutures from their invention, patents, commercialization to clinical applications. It is very useful summary for both surgeons and manufactures to further improve and increase their clinical application to benefit for both patients and surgeons. I recommend to accept in current form.

Two minor comments:

Line 196, There is extra period”.” after that which should be deleted.

Lines 551-553, fonts and size are different. 

Author Response

Thank you for providing your valuable comments for the review manuscript.

The extra period in the document has been deleted. 

The font and size of the text have been modified to match the text style to the rest of the manuscript.

Reviewer 5 Report

In this manuscript Authors presented Barbed Sutures evolution. The authors have not presented this concept very well. There is no recent update about this topic.

Update the recent references, preferable recent 5 years research. At present very less references. In this topic, many recent references available.

Section should be number in the better way. Foe ex: introduction should be 1, history should be 2 and like this…

How Barbed Sutures improve its solubility after surgery. Need to give insight in this review.

Is it harmful for the human body? Consider its toxicity to the human. Consider also its cost than other surgical tools.

How Barbed Sutures is helpful to the human, give its quantitative idea including its proper mechanism. Add some recent studies about this.

References number should be added align with the text not in superscript mode.  

Insert reference in Line 47.

In introduction section there are very few references and information please improve it.

Give some statistical analysis of the research related to this study.

Figures quality is very low; please improve it as per the journal guidelines.

Line 637, how discussion section comes for a review manuscript?

No future direction of this manuscript. Make a separate section.

Author Response

Thank you for providing your valuable comments on the manuscript.

We have revised and edited the English language to the best of our ability. 

In this manuscript Authors presented Barbed Sutures evolution. The authors have not presented this concept very well. There is no recent update about this topic.

Update the recent references, preferable recent 5 years research. At present very less references. In this topic, many recent references available.

We have searched for most recent references and also included some of the recent references that we had missed earlier and we have already listed most of the references from the recent years. When we looked into other references available online, even though there were recent references, they were all looking into the clinical aspect and not on the barbed sutures concept and so we had to leave those references out from the reference list.

Section should be number in the better way. Foe ex: introduction should be 1, history should be 2 and like this…

We have re-numbered the sections in a better way for easy perception of the message and the body of the manuscript.

How Barbed Sutures improve its solubility after surgery. Need to give insight in this review.

The barbed sutures does not have an impact on the degradation of the suture as the barbed sutures are usually manufactured using the commercially available suture monofilaments. And barbed sutures are manufactured just by creating the projections using a mechanical blade assembly and so there is no change to the solubility after production of barbed sutures.

Is it harmful for the human body? Consider its toxicity to the human. Consider also its cost than other surgical tools.

As these barbed sutures are produced using the available suture monofilaments which have been widely used for different surgical wound closure procedures and so there is no modification to the material properties after barbing of sutures and so they don't exhibit any major toxic reactions from the tissues. And related to the cost of these surgical tools, even though these barbed sutures are expensive as mentioned in the manuscript they have benefits and lower complications.

How Barbed Sutures is helpful to the human, give its quantitative idea including its proper mechanism. Add some recent studies about this.

The beneficial use of barbed sutures during different surgical procedures and how it helps in reducing the operating stress on the surgeons who perform the procedures are mentioned in section 3.2 where the role of these barbed sutures and their advantages of the use of these specific type of sutures were mentioned. 

References number should be added align with the text not in superscript mode. 

The reference style was ACS citation style which was provided by the journal. That is the reason why we have referenced the citations in this way.

Insert reference in Line 47.

We have added the reference for the line 47.

In introduction section there are very few references and information please improve it.

Additional references have been added to the introduction section.

Give some statistical analysis of the research related to this study.

It would be very difficult to include statistical analysis as in section 3.2, the studies are from different clinical reviews and they were all based on clinical experiences by surgeons. Since the review consists of different surgical procedures involving different barbed sutures, it would be difficult to detail in a statistical form of the data and outcomes mentioned in the different studies. 

Figures quality is very low; please improve it as per the journal guidelines.

We have made the figures with the best quality possible and also made sure that they abide by the journal guidelines as well.

Line 637, how discussion section comes for a review manuscript?

The section was mentioned mistakenly as discussion section instead of as a part of conclusion. We have made the edit as to clearly mention it as a part of conclusion section.

No future direction of this manuscript. Make a separate section.

We have made the future directions and conclusions as a separate section at the end of the manuscript.

Round 2

Reviewer 5 Report

The authors improved the manuscript as per my suggestions. 

Author Response

Thank you for your valuable comments to our revised manuscript.